# Effects of Intraoperative Opioid Use and a Combined Anesthesia Protocol in Patients Undergoing Radical Cystectomy for Urothelial Carcinoma of the Bladder—A Single-Center Experience

**DOI:** 10.3390/cancers16193411

**Published:** 2024-10-08

**Authors:** Julian Marcon, Fatima Yefsah, Gerald B. Schulz, Philipp Weinhold, Severin Rodler, Lennert Eismann, Yannic Volz, Paulo L. Pfitzinger, Christian G. Stief, Christian Kowalski, Daniel Siegl, Alexander Buchner, Nikolaos Pyrgidis, Jan-Friedrich Jokisch

**Affiliations:** 1Department of Urology, University Hospital of the LMU Munich, 80336 Munich, Germany; julian.marcon@med.uni-muenchen.de (J.M.); fatima93@web.de (F.Y.); gerald.schulz@med.uni-muenchen.de (G.B.S.); philipp.weinhold@med.uni-muenchen.de (P.W.); severin.rodler@med.uni-muenchen.de (S.R.); lennert.eismann@med.uni-muenchen.de (L.E.); yannic.volz@med.uni-muenchen.de (Y.V.); paulo.pfitzinger@med.uni-muenchen.de (P.L.P.); christian.stief@med.uni-muenchen.de (C.G.S.); alexander.buchner@med.uni-muenchen.de (A.B.); friedrich.jokisch@med.uni-muenchen.de (J.-F.J.); 2Department of Urology, University Hospital Schleswig-Holstein, 24105 Kiel, Germany; 3Department of Anesthesiology, University Hospital of the LMU Munich, 80336 Munich, Germany; christian.kowalski@med.uni-muenchen.de (C.K.); daniel.siegl@med.uni-muenchen.de (D.S.)

**Keywords:** analgesics, opioid, carcinoma, transitional cell, cystectomy, analgesia, epidural, urologic surgical procedures

## Abstract

**Simple Summary:**

This study evaluated the impact of intraoperative opioid use and anesthesia type on outcomes for patients undergoing radical cystectomy for bladder cancer. Data from 508 patients treated between 2015 and 2022 were analyzed. Most (82%) received combined intravenous and epidural anesthesia, while the rest received intravenous-only anesthesia. Results showed that combined anesthesia was linked to better overall survival and fewer intensive care unit admissions. However, opioid dosage and type did not significantly affect survival, recurrence rates, or major perioperative outcomes. The findings are limited by the study’s single-center, retrospective nature, and further research is needed to confirm the safety of opioids in patients undergoing radical cystectomy.

**Abstract:**

Background: An increased intraoperative opioid dose seems to lead to worse outcomes in several types of cancer. We assessed the effect of intraoperatively administered opioids as well as the type of anesthesia on survival, recurrence rates and major perioperative outcomes in patients who underwent radical cystectomy (RC) for urothelial carcinoma of the urinary bladder. Methods: We included patients who underwent open RC at our center between 2015 and 2022. The role of the type and dosage of intraoperative opioid agents, such as remifentanil, sufentanil and morphine milligram equivalents (MME), as well as the type of anesthesia (intravenous only versus intravenous/epidural), was assessed regarding perioperative and long-term outcomes after RC. Results: A total of 508 patients with a median age of 73 years (IQR: 64–78) were included. Overall, 92 (18%) patients received intravenous anesthesia, whereas 416 (82%) received combined anesthesia. At a median follow-up of 270 days (IQR: 98–808), 108 (21%) deaths and 106 (21%) recurrences occurred. Combined anesthesia was associated with better survival (HR:0.63, 95% CI: 0.4–0.97, *p* = 0.037) and lower intensive care unit admission rates (OR: 0.49, 95% CI: 0.31–0.77, *p* = 0.002) in the univariate analysis (unadjusted). The type and dosage of intraoperative opioid agents did not affect long-term survival and recurrence rates, as well as major perioperative outcomes. Nevertheless, the findings of our study were limited by its single-center, retrospective design. Conclusion: The use of intraoperative opioids was not associated with worse outcomes in our cohort, while the use of additional epidural anesthesia seems to be beneficial in terms of overall survival and intensive care unit admissions. Nevertheless, further research is mandatory to validate the safety of opioids in patients undergoing RC.

## 1. Introduction

In patients with muscle-invasive or very high-risk non-muscle-invasive urothelial carcinoma of the bladder, radical cystectomy (RC) with pelvic lymph node dissection and urinary diversion, with or without neoadjuvant systemic therapy, represents the gold standard in surgical treatment [1]. Although improving inpatient care and surgical technique has led to better patient outcomes, RC is still associated with relevant perioperative morbidity and mortality [2]. Approximately 20% of perioperative complications are potentially lethal [3,4]. The rate of 10-year recurrence-free survival (RFS) after open RC is about 63%, while the rate of overall survival (OS) after ten years is approximately 46% [5], with several factors, such as locally advanced disease, negatively impacting long-term survival [6].

The use of opioids is the mainstay in modern analgesia during intraoperative anesthesia [7]. Several studies evaluating the risk of perioperative opioid use in several cancers, such as in renal cell carcinoma [8], prostate cancer [9] and laryngeal squamous cell carcinoma [10], have associated higher doses with worse oncological outcomes. On the contrary, other studies have observed no unfavorable effect in patients undergoing surgery for primary liver cancer [11] or breast cancer [12]. With respect to the pathophysiological background of the potential negative influence of opioids, authors postulated a possible immunomodulatory effect of µ-opioid receptors (MORs) expressed on tumor cells with a consecutive promoting impact on tumor angiogenesis and a negative effect on immune response through immunosuppression [13]. A prior study on patients undergoing RC for urothelial carcinoma suggests a negative effect of higher opioid doses during anesthesia on recurrence-free and cancer-specific survival in a group receiving epidural anesthesia [14]. However, data regarding intraoperative opioid use in this tumor entity are scarce and often contradictory. Given the discordance in the existing literature, it seems necessary to investigate the potential risks associated with intraoperative opioid use during RC further [15].

With the present study, we aimed to assess the risk of higher intraoperative opioid doses, with a focus on modern substances remifentanil and sufentanil, as well as the impact of combined intravenous and epidural anesthesia during RC on overall survival, recurrence rates and postoperative complications in patients with urothelial carcinoma of the bladder in a high-volume tertiary urology care center.

## 2. Material and Methods

### 2.1. Study Design

After obtaining approval from our internal review board, we performed a retrospective analysis from our prospectively maintained database for patients who underwent open RC between 2015 and 2022. The present study conforms with the Declaration of Helsinki, and all findings are presented based on the STROBE statement [16]. All patients provided written informed consent upon inclusion. For the present analysis, we included patients diagnosed with pure urothelial carcinoma of the urinary bladder undergoing RC with neoadjuvant or adjuvant chemotherapy. Moreover, only patients with complete demographic data, full documentation of intraoperative opioid doses, and comprehensive follow-up information were included in the study. On the contrary, we excluded all patients with missing data, as well as those undergoing RC for variant or mixed histology or for non-oncological reasons.

### 2.2. Anesthesia Protocol

Remifentanil or sufentanil were used in all cases as intraoperative opioid agents during RC. Their total intraoperative dosage was computed. For the purposes of the analysis, intraoperative remifentanil and sufentanil were converted into oral morphine milligram equivalents (MME), measured as a continuous variable, and presented per 10 MME. Based on the available literature, 10 MME was considered the equivalent of 50 μg of i.v. remifentanil, which has approximately the same therapeutic potency as fentanyl, and 5 μg of i.v. sufentanil, which is about ten times as potent as fentanyl [17,18]. Furthermore, a subdivision of patients according to the total dose of remifentanil, sufentanil and 10 MME was created based on the 75th percentile according to clinical judgment. In particular, patients were grouped into low- (<75th percentile) and high- (≥75th percentile) dose recipients of remifentanil, sufentanil and 10 MME in an attempt to objectively assess the effect of high opioid dosages on oncological outcomes. In patients who underwent combined epidural and intravenous anesthesia, an epidural catheter was placed at the discretion of the surgeon and anesthesiologist between levels T12 and L2. Sufentanil was the only opioid substance applied for epidural anesthesia with a standardized approach based on relevant anesthesiological protocols. Combined intravenous and epidural anesthesia was offered to all eligible patients by the anesthesiologists aiming to share decision-making for the type of anesthesia.

### 2.3. Outcomes and Statistical Analysis

The primary outcome of the present study was to assess the role of major intraoperative opioid agents in predicting perioperative outcomes, as well as long-term overall survival and recurrence rates after RC for urothelial cancer of the urinary bladder. Secondary outcomes included the role of the intraoperative opioid dosage in predicting these outcomes. Remifentanil, sufentanil and 10 MME were examined separately with regard to the outcomes of interest. Moreover, the role of the type of anesthesia (total intravenous anesthesia only versus intravenous + additional epidural anesthesia) was also evaluated.

All continuous variables were presented as medians with interquartile ranges (IQRs) and were compared with the Mann–Whitney U test. Similarly, all categorical variables were presented as frequencies with proportions and were compared with the chi-squared test. A univariate regression analysis was performed to assess the effect of opioid and type of anesthesia on perioperative and long-term outcomes. Survival estimates were computed using Cox regression analysis and a log-rank test. For all analyses, hazard ratios (HRs) or odds ratios (ORs) with the corresponding 95% confidence intervals (CIs) were estimated. A two-sided *p*-value < 0.05 was considered statistically significant for all analyses, which were performed with the R statistical software (version 3.6.3, R Core Team 2020).

## 3. Results

### 3.1. Baseline Characteristics

A total of 508 patients underwent open RC from 2015 to 2022 at our institution for urothelial cancer and presented full documentation of intraoperative opioid doses, as well as comprehensive follow-up information. Their median age was 73 years (IQR: 64–78), their median BMI was 26 kg/m^2^ (IQR: 24–28) and 383 (75%) patients were male. A total of 250 (61%) patients were smokers, 280 (56%) had hypertension and 172 (34%) had diabetes. Ileal conduit was the most preferred urinary diversion with 292 (57%) cases, followed by orthotopic ileal neobladder with 203 (40%) cases. The median operative time of RC was 223 minutes (IQR: 186–265). Of note, 342 (69%) patients presented locally advanced bladder cancer (≥pT3), 120 (27%) positive lymph nodes (pN+) and 68 (14%) positive surgical margins during open RC.

Overall, 92 (18%) patients received only intravenous anesthesia, whereas 416 (82%) received combined intravenous and epidural anesthesia. The median sufentanil dosage was 10 µg (IQR: 10–30), the median remifentanil dosage was 2317 µg (IQR: 1380–3565) and the median 10 MME dosage was 7 µg (IQR: 2–89). A high dosage (≥75th percentile) of sufentanil was defined as 30 µg, a high dosage (≥75th percentile) of remifentanil was defined as 3565 µg and a high dosage (≥75th percentile) of 10 MME was defined as 89 µg. Patients receiving only intravenous anesthesia did not present statistically significant differences in the type and dosage of intraoperative opioid agents compared to patients receiving combined intravenous and epidural anesthesia. The baseline characteristics of the whole study cohort and the comparisons based on the type of anesthesia are presented in Table 1.

### 3.2. Overall Survival and Recurrence Rates

At a median follow-up of 270 days (IQR: 98–808), 108 (21%) deaths occurred. Of them, 26 (28%) were reported in patients undergoing only intravenous anesthesia, and 82 (20%) were reported in those undergoing combined intravenous and epidural anesthesia. Combined intravenous and epidural anesthesia was associated with better survival (log-rank test: *p* = 0.036). The corresponding Kaplan–Meier curve is depicted in Figure 1. In the univariate Cox regression analysis, combined intravenous and epidural anesthesia was associated with better survival (HR: 0.63, 95% CI: 0.4 to 0.97, *p* = 0.037) compared to solely intravenous anesthesia. On the contrary, patients receiving higher intraoperative doses of remifentanil (*p* = 0.4), sufentanil (*p* = 0.7) or MME (*p* = 0.8) did not present worse survival outcomes.

During follow-up, 106 (28%) recurrences of urothelial cancer occurred. Of them, 16 (25%) were reported in patients undergoing only intravenous anesthesia, and 82 (20%) were reported in those undergoing combined intravenous and epidural anesthesia. The type of anesthesia (*p* = 0.7), as well as the use of higher intraoperative doses of remifentanil (*p* = 0.3), sufentanil (*p* = 0.9) or MME (*p* = 0.07), did not affect the recurrences of urothelial cancer. The corresponding univariate Cox regression analyses for the time to death and the time to recurrence of urothelial cancer after RC are available in Table 2.

### 3.3. Blood Transfusions, Severe Clavien–Dindo Complications, and Intensive Care Unit (ICU) Admissions

During their hospital stays, 106 (21%) patients required intraoperative allogeneic blood transfusions, 64 (14%) patients developed a severe Clavien–Dindo complication (at least grade 3) and 188 (37%) patients were admitted to the ICU. All Clavien–Dindo complications were defined based on the validated criteria for urological operations [19]. Moreover, the need for admission to the ICU was based on clinical judgment according to the preoperative patient characteristics as well as the intraoperative findings. The type of anesthesia (*p* = 0.1 and *p* = 0.6), as well as the use of higher intraoperative doses of remifentanil (*p* = 0.8 and *p* > 0.9), sufentanil (*p* = 0.6 and *p* > 0.9) or MME (*p* = 0.7 and *p* = 0.7), did not affect the transfusion rates and the development of severe Clavien–Dindo complications. On the contrary, 47 (51%) patients were admitted to the ICU after undergoing only intravenous anesthesia compared to 141 (34%) patients who underwent combined intravenous and epidural anesthesia. In the univariate logistic regression analysis, combined intravenous and epidural anesthesia was associated with lower rates of admission to the ICU (OR: 0.49, 95% CI: 0.31 to 0.77, *p* = 0.002). On the contrary, patients receiving higher intraoperative doses of remifentanil (*p* = 0.3), sufentanil (*p* = 0.8) or MME (*p* = 0.3) did not present higher rates of admission to the ICU. The corresponding univariate logistic regression analyses for transfusion rates, perioperative severe Clavien–Dindo complications and admissions to the ICU after RC are displayed in Table 3.

### 3.4. Length of Hospital Stay and Blood Loss

The median length of hospital stay after RC was 19 days (IQR: 17–22) and the median intraoperative blood loss was 400 ml (IQR: 200–700). The type of anesthesia (*p* = 0.7 and *p* = 0.9), as well as the use of higher intraoperative doses of remifentanil (*p* = 0.054 and *p* = 0.2), sufentanil (*p* = 0.6 and *p* = 0.3) or MME (*p* > 0.9 and *p* = 0.2), did not affect the length of hospital stay and the median intraoperative blood loss. The corresponding univariate linear regression analyses for the length of hospital stay and the intraoperative blood loss after RC are summarized in Table 4.

## 4. Discussion

The findings of the present cohort study in patients undergoing RC for urothelial cancer indicate that the type and the dosage of intraoperative opioid agents did not affect long-term survival and recurrence rates. Similarly, the type and the dosage of intraoperative opioid agents also did not affect major perioperative outcomes such as transfusion rates, perioperative severe Clavien–Dindo complications, ICU admissions, length of hospital stay and intraoperative blood loss. On the contrary, it seems that the combination of intravenous and epidural anesthesia may lead to better overall survival and to lower rates of ICU admissions compared to solely intravenous anesthesia. More specifically, in the univariate regression analysis, the combination of intravenous and epidural anesthesia was associated with 59% better overall survival and with 100% lower ICU admissions compared to solely intravenous anesthesia. The latter might be attributed to better perioperative pain control and improved respiratory function, leading, in turn, to better hemodynamic stability and a reduction in systemic inflammatory response.

Opioids are an integral part of both perioperative anesthesia and pain management in cancer patients. These substances bind to a variety of mu-opioid receptors (MORs) to take analgesic effect. In esophageal cancer, a greater expression of MORs is associated with more advanced disease [20]. Opioid substances have been shown to have immunosuppressive properties. Preclinical studies report the cancer-promoting effects of opioids, which have mostly been regarding morphine, and have linked them to the promotion of direct cancer cell growth, the advancement of metastatic disease and immunosuppressive features [21,22,23]. Conversely, other animal studies have found morphine to also possess antitumor effects [24]. A potential negative impact of perioperative opioid administration may be caused by a combination of a perioperative catecholamine surge as well as immunosuppressive and proangiogenic effects [25]. A negative impact of intraoperatively applied opioids has been observed in multiple studies on different cancer types. With respect to genitourinary cancers, a previous study evaluated the prognostic impact of intraoperative opioid dose in more than 2000 patients undergoing partial or radical nephrectomy for renal cell carcinoma. The authors found a lower recurrence-free survival rate in patients in whom higher opioid doses, measured by units of 10 MME, as used in the present study, were administered [8]. A prior study assessed the effect of intraoperative opioid dose in a cohort of 439 patients who underwent RC for urothelial carcinoma. The authors found worse cancer-specific and recurrence-free survival in patients with combined epidural and intravenous anesthesia. Interestingly, patients who underwent epidural anesthesia received higher opioid doses, which was not the case in our cohort. Also, in the previous study, sufentanil was the only opioid substance applied, while in our study, patients received both remifentanil and sufentanil [14]. Whereas no propensity score matching was used in the present study, there was no significant difference between previously used parameters such as TNM stage or age between both anesthesia groups.

Regarding combined anesthesia using both intravenously and epidurally administered opioids, a beneficial effect on overall survival could be determined for colon and ovarian cancer in two recent meta-analyses [26,27]. These data are contrary to the findings of the group of Chipollini et al. but are in accordance with our present data with a significant benefit on overall survival [14]. In particular, Chipollini et al. [14] indicated that epidural anesthesia with sufentanil was associated with worse recurrence and disease-free survival due to the increased total morphine equivalents that the included patient received. Still, in the analysis of low- (<75th percentile) and high- (≥75th percentile) dose recipients of remifentanil, sufentanil and 10 MME that we performed, no significant differences were observed in our tertiary reference center. With regard to the difference in oncological outcomes depending on intraoperative opioid doses observed in a variety of studies, it must be noted that there is significant heterogeneity in the reported cohorts as well as opioids used during anesthesia. The different results may furthermore be caused by a distinct effect of intraoperative opioids depending on the underlying cancer type and associated tumor genetics.

It should be highlighted that in this cohort study, the overall perioperative complication rates were relatively low. The latter may be attributed to the large number of RCs performed yearly in our center [28]. Indeed, increased annual hospital volume is associated with improved perioperative outcomes for RC [29]. High-volume centers not only provide the required infrastructure and well-trained operative teams to prevent perioperative complications but also provide specialized anesthesiology teams that can optimize the intraoperative and short-term postoperative management of patients [30]. By applying optimal anesthetic techniques, implementing an effective multimodal pain management protocol, optimizing fluid administration, ensuring perioperative rigorous monitoring and using Enhanced Recovery After Surgery (ERAS) protocols, experienced anesthesiology teams can achieve optimal perioperative outcomes [31]. Based on the previous notion, in our study, the correct use of intravenous and epidural opioids was associated with beneficial outcomes not only in terms of perioperative outcomes but also in terms of long-term mortality. Furthermore, it is noteworthy that the relatively high rate of deaths (21%) and recurrences (28%) during follow-up is presumably due to the high percentage of patients with locally advanced tumors (69%) in our cohort.

Furthermore, it needs to be stated that the findings of the present study were mitigated by some limitations relevant to its single-center design, relatively short follow-up period, potential selection bias and lack of randomization. Based on the previous notion, the possibility of unmeasured confounders influencing the results, such as variations in surgical technique or perioperative care, could not be addressed. It should be highlighted that it was beyond the scope of the present study to assess for the effect of the type of the urinary diversion as well as the surgeon’s caseload since multiple studies on the matter have been previously published [32,33]. All analyses were solely performed on patients with urothelial cancer, excluding all variant histologies, of the urinary bladder. Accordingly, the role of other intraoperative or perioperative analgesics such as local anesthetics, non-steroid anti-inflammatory drugs or metamizole on short- and long-term outcomes was not explored. Importantly, we could not assess further complications such as ileus, sepsis or reintervention rates. Finally, given the low rate of complications, deaths and recurrences, multivariable regression analyses could not be performed.

## 5. Conclusions

Intraoperative opioid application was safe to use during open radical cystectomy, with a suggested prognostic benefit of a combined intravenous and epidural anesthesia protocol. Nevertheless, the observational nature of our study warrants a cautious interpretation of its findings. Therefore, further multi-center studies would be desirable to assess this observation on a larger scale. Accordingly, prospective, multicenter studies or randomized controlled trials to confirm our findings are mandatory.

## Figures and Tables

**Figure 1 cancers-16-03411-f001:**
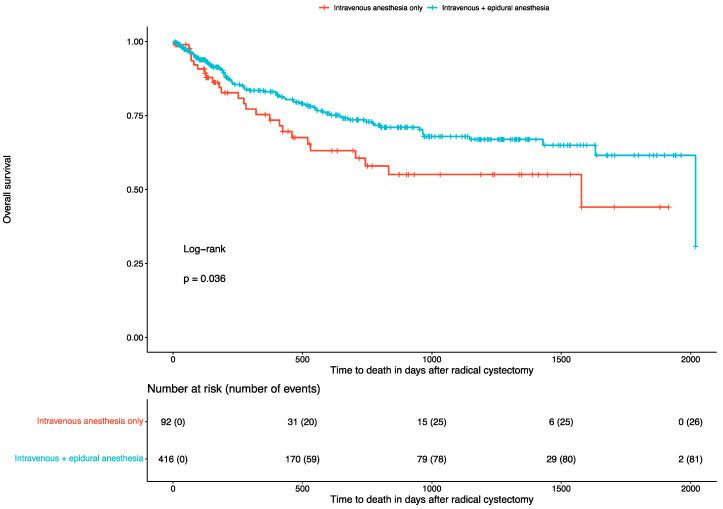
Kaplan–Meier curves for overall survival in patients undergoing radical cystectomy.

**Table 1 cancers-16-03411-t001:** Baseline characteristics of the included patients undergoing radical cystectomy based on the type of anesthesia.

Characteristic	Overall, *n* = 508	Intravenous Anesthesia Only, *n* = 92	Intravenous + Epidural Anesthesia, *n* = 416	*p*-Value
Age (years)	73 (64–78)	73 (65–78)	73 (64–78)	>0.9
Males	383 (75%)	70 (76%)	313 (75%)	>0.9
BMI (kg/m^2^)	26 (24–28)	27 (23–28)	26 (24–28)	>0.9
Smokers	250 (61%)	37 (48%)	213 (65%)	**0.01**
Alcohol consumption	110 (32%)	18 (27%)	92 (33%)	0.5
Heart disease	159 (32%)	38 (41%)	121 (30%)	**0.04**
Hypertension	280 (56%)	54 (59%)	226 (55%)	0.6
Diabetes	172 (34%)	29 (32%)	143 (35%)	0.6
ASA				**0.04**
1	4 (0.8%)	0 (0%)	4 (1.0%)	
2	90 (18%)	17 (19%)	73 (18%)	
3	402 (79%)	68 (76%)	334 (80%)	
4	10 (2.0%)	5 (5.6%)	5 (1.2%)	
Urinary diversion				0.3
Ileal conduit	292 (57%)	61 (66%)	231 (56%)	
Neobladder	203 (40%)	29 (32%)	174 (42%)	
Pouch	5 (1.0%)	1 (1.1%)	4 (1.0%)	
Ureterocutaneostomy	8 (1.6%)	1 (1.1%)	7 (1.7%)	
Operative time (min)	223 (186–265)	224 (192–269)	223 (186–265)	0.9
Blood loss (mL)	400 (200–700)	400 (200–700)	400 (200–700)	0.5
T after cystectomy				>0.9
≤T2	156 (31%)	28 (30%)	128 (32%)	
≥T3	342 (69%)	64 (70%)	278 (68%)	
Positive lymph nodes	120 (27%)	25 (31%)	95 (26%)	0.4
Positive surgical margins	68 (14%)	12 (13%)	56 (14%)	>0.9
Hospital stay (days)	19 (16–22)	18 (16–21)	19 (16–22)	0.5
Perioperative chemotherapy	184 (36%)	27 (29%)	157 (38%)	0.2
Clavien–Dindo complications				0.2
No	273 (61%)	43 (54%)	230 (62%)	
1	5 (1.1%)	3 (3.8%)	2 (0.5%)	
2	107 (24%)	21 (26%)	86 (23%)	
3	28 (6.2%)	5 (6.3%)	23 (6.2%)	
4	30 (6.7%)	7 (8.8%)	23 (6.2%)	
5	6 (1.3%)	1 (1.3%)	5 (1.4%)	
Allogeneic blood transfusion	106 (21%)	25 (27%)	81 (20%)	0.14
Admission to ICU	188 (37%)	47 (51%)	141 (34%)	**0.003**
Sufentanil (µg)	10 (10–30)	10 (8–20)	10 (10–30)	0.08
Remifentanil (µg)	2317 (1380–3565)	3112 (1272–4277)	2243 (1408–3367)	0.08
10 MME	7 (2–89)	8 (2–74)	7 (2–89)	0.5

Values are presented as median (interquartile range) or *n* (%). Bold *p*-values indicate statistically significant values. ASA: American Society of Anesthesiology, BMI: Body Mass Index, ICU: intensive care unit, MME: morphine milligram equivalents.

**Table 2 cancers-16-03411-t002:** Univariate Cox regression analysis for overall survival and urothelial cancer recurrence in patients undergoing radical cystectomy.

	Outcome	Univariate Cox Regression		
		HR	95% CI	*p*-Value
Mortality	Type of anesthesia			
	Intravenous anesthesia only	—	—	
	Intravenous + epidural anesthesia	0.63	0.40, 0.97	**0.037**
	High dose of remifentanil	0.76	0.39, 1.46	0.4
	High dose of sufentanil	1.08	0.71, 1.64	0.7
	High dose of MME	0.96	0.63, 1.47	0.8
	Epidural anesthesia	0.67	0.45, 1.02	0.059
Recurrence	Type of anesthesia			
	Intravenous anesthesia only	—	—	
	Intravenous + epidural anesthesia	1.10	0.64, 1.87	0.7
	High dose of remifentanil	1.36	0.78, 2.36	0.3
	High dose of sufentanil	1.04	0.68, 1.60	0.9
	High dose of MME	1.44	0.94, 2.1	0.07
	Epidural anesthesia	1.18	0.74, 1.89	0.5

Bold *p*-values indicate statistically significant values. CI: confidence interval, HR: hazard ratio, MME: morphine milligram equivalents.

**Table 3 cancers-16-03411-t003:** Univariate logistic regression analysis for transfusion rates, perioperative severe Clavien–Dindo complications and admissions to the intensive care unit in patients undergoing radical cystectomy.

	Outcome	Univariate Logistic Regression		
		OR	95% CI	*p*-Value
Transfusion	Type of anesthesia			
	Intravenous anesthesia only	—	—	
	Intravenous + epidural anesthesia	0.65	0.39, 1.11	0.10
	High dose of remifentanil	1.08	0.51, 2.19	0.8
	High dose of sufentanil	1.15	0.70, 1.84	0.6
	High dose of MME	1.09	0.66, 1.76	0.7
	Epidural anesthesia	0.62	0.39, 1.01	0.052
Severe Clavien–Dindo complications	Type of anesthesia			
	Intravenous anesthesia only	—	—	
	Intravenous + epidural anesthesia	0.83	0.44, 1.66	0.6
	High dose of remifentanil	0.98	0.34, 2.50	>0.9
	High dose of sufentanil	1.04	0.55, 1.87	>0.9
	High dose of MME	1.14	0.61, 2.05	0.7
	Epidural anesthesia	0.68	0.38, 1.25	0.2
Admission to intensive care unit	Type of anesthesia			
	Intravenous anesthesia only	—	—	
	Intravenous + epidural anesthesia	0.49	0.31, 0.77	**0.002**
	High dose of remifentanil	0.73	0.38, 1.37	0.3
	High dose of sufentanil	1.05	0.70, 1.58	0.8
	High dose of MME	1.24	0.82, 1.87	0.3
	Epidural anesthesia	0.61	0.4, 0.93	**0.02**

Bold *p*-values indicate statistically significant values. CI: confidence interval, OR: odds ratio, MME: morphine milligram equivalents.

**Table 4 cancers-16-03411-t004:** Univariate linear regression analysis for length of hospital stay and intraoperative blood loss in patients undergoing radical cystectomy.

	Outcome	Univariate Linear Regression		
		Beta	95% CI	*p*-Value
Length of hospital stay (days)	Type of anesthesia			
	Intravenous anesthesia only	—	—	
	Intravenous + epidural anesthesia	−0.40	−2.3, 1.5	0.7
	High dose of remifentanil	2.3	−0.04, 4.6	0.054
	High dose of sufentanil	0.45	−1.1, 2.0	0.6
	High dose of MME	−0.05	−1.7, 1.6	>0.9
	Epidural anesthesia	−0.41	−2.1, 1.3	0.6
Blood loss (mL)	Type of anesthesia			
	Intravenous anesthesia only	—	—	
	Intravenous + epidural anesthesia	−7.8	−119, 103	0.9
	High dose of remifentanil	82	−41, 205	0.2
	High dose of sufentanil	51	−48, 149	0.3
	High dose of MME	65	−33, 164	0.2
	Epidural anesthesia	21	−81, 122	0.7

CI: confidence interval, MME: morphine milligram equivalents.

## Data Availability

All data generated or analyzed during this study are included in this article. Further inquiries can be directed to the corresponding author.

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
