# Peer review of "Effects of Intraoperative Opioid Use and a Combined Anesthesia Protocol in Patients Undergoing Radical Cystectomy for Urothelial Carcinoma of the Bladder—A Single-Center Experience"

_cancers, 2024, doi:10.3390/cancers16193411_

Round 1

Reviewer 1 Report

Comments and Suggestions for Authors

Effects of intraoperative opioid use and a combined anesthesia protocol in patients undergoing radical cystectomy for urothelial carcinoma of the bladder – a single-center experience

General Comments:

This manuscript provides a retrospective cohort study assessing the effect of intraoperative opioid dosage and type of anesthesia on perioperative and long-term outcomes in patients undergoing radical cystectomy (RC) for urothelial carcinoma of the bladder. Results demonstrate that intraoperative opioids did not adversely impact survival rates or recurrence rates; combined intravenous and epidural anesthesia proved more successful at improving overall survival with decreased ICU admissions and ICU stays.

The topic is clinically significant, as understanding the impact of anesthetic management on oncological outcomes could inform perioperative practices. Overall, the manuscript is well structured and written; however, there may be several methodological issues or areas needing clarification that need addressing in order to strengthen validity and impact of this research study.

Specific Comments:

Abstract:

- Clarity: Abstract is brief yet could more effectively reflect limitations of study. Consider adding a sentence acknowledging retrodesign and calling for further research to ensure accuracy.

- Results Presentation: Note the nature of any Hazard Ratios (HRs) or Odds Ratios (ORs) reported. For example, adjusted or unadjusted should be stated accordingly.

Introduction:

- Background Rationale: The introduction provides an effective overview of RC and opioids' potential impact on cancer outcomes, but could use additional detail about how opioids might influence tumor progression; such as through immunosuppression or angiogenesis.

- Literature Context: Exploration of discrepant findings from previous studies regarding intraoperative opioid usage for various cancer types will help establish a clearer rationale for this current research study.

- Aim Clarification: The study objectives have been stated, but could be further refined. Be clear as to whether your focus lies with overall survival, recurrence rates, postoperative complications or all these outcomes.

Methods:

- Study Design:

  - Retrospective Nature: At the start of your Methods section, clearly state that this study is retrospective cohort in nature. 

- Patient Selection:

  - Inclusion/Exclusion Criteria: Provide specific inclusion and exclusion criteria. For instance, include information regarding how mixed histologies were managed as well as whether patients receiving either neoadjuvant or adjuvant chemotherapy are considered.

  - Baseline Differences: Explain how any possible baseline differences were addressed between anesthesia groups.

- Anesthesia Protocol:

  - Decision Process: Provide details regarding how the decision between intravenous only or combined intravenous and epidural anesthesia was reached; was it determined based on patient preferences, surgeon preferences or anesthesiologist discretion?

  - Standardization: Indicate whether dosing of epidural sufentanil was standard across patients.

  - Opioid Conversion: Provide more detail on how remifentanil and sufentanil doses were converted to morphine milligram equivalents (MMEs), including any references or formulae used.

- Outcomes and Definitions:

  - Primary and Secondary Outcomes: Define both primary and secondary outcomes clearly. For instance, outline whether overall survival refers to all-cause mortality or cancer-specific mortality.

  - Perioperative Outcomes: Define what constitutes a serious Clavien-Dindo complication and how ICU admission decisions were determined (e.g., predefined criteria or clinician judgment).

- Statistical Analysis:

  - Adjustment for Confounders: Detail any methods used to adjust for potential confounding variables, such as multivariable regression analyses.

  - Rationale for Cut-offs: Justify the choice of the 75th percentile as the cut-off for high versus low opioid dosage groups.

  - Handling of Missing Data: Explain how missing data were addressed, if applicable.

  - Software Version: Update the statistical software version if more recent than R 3.6.3 is available, or justify using this version.

Results:

- Baseline Characteristics:

  - Comparative Analysis: Provide p-values for all comparisons in Table 1 to highlight any significant differences between the anesthesia groups.

  - Additional Variables: Consider including other relevant variables, such as ASA scores, comorbidities, or tumor characteristics, and discuss any significant differences.

- Survival and Recurrence Outcomes:

  - Follow-up Duration: The median follow-up of 270 days is relatively short for oncological outcomes. Acknowledge this limitation in both the Results and Discussion sections.

  - Multivariable Analysis: Given the number of events (108 deaths and 106 recurrences), it seems feasible to perform multivariable analyses to adjust for confounders. Explain why this was not done or include these analyses.

  - Figures: Ensure that Figure 1 (Kaplan-Meier curve) includes clearly labeled axes, numbers at risk, and confidence intervals if possible.

- Perioperative Outcomes:

  - ICU Admissions: Discuss possible reasons why combined anesthesia was associated with lower ICU admission rates.

  - Complication Rates: Provide more detail on the types of complications observed and whether they differed between groups.

Discussion:

- Interpretation of Findings:

  - Mechanisms: Explore potential mechanisms for the observed association between combined anesthesia and improved survival and reduced ICU admissions.

  - Comparison with Literature: Expand the discussion to compare and contrast your findings with those of previous studies, such as the study by Chipollini et al. mentioned in the text.

- Limitations:

  - Acknowledgment: Discuss the retrospective design, potential selection bias, lack of randomization, and the short follow-up period as limitations.

  - Confounding Factors: Address the possibility of unmeasured confounders influencing the results, such as variations in surgical technique or perioperative care.

- Recommendations:

  - Future Research: Suggest the need for prospective, multicenter studies or randomized controlled trials to confirm these findings.

Conclusion:

- Balanced Summary: The conclusion should reflect both the findings and the limitations of the study. Emphasize that while intraoperative opioid use did not adversely affect outcomes, the observational nature of the study warrants cautious interpretation.

Additional Comments:

- Ethical Considerations:

  - Ethics Statement: The ethics statement is contradictory. It mentions approval by the Ethics Committee but also states that the article does not contain any studies with human participants. Clarify this section to accurately reflect the ethical approval and consent processes.

- Data Availability:

  - Transparency: If possible, provide information on how the dataset can be accessed for verification or further research, respecting patient confidentiality. 

Recommendation:

Revisions Required

The manuscript addresses an important clinical question but requires significant revisions to enhance clarity, address methodological concerns, and provide a more robust analysis of the data. I recommend the authors address the above comments, particularly focusing on methodological transparency, statistical analyses, and thorough discussion of the findings within the context of existing literature.

Comments on the Quality of English Language

  - Grammar and Syntax: There are minor grammatical errors and typos throughout the manuscript. A thorough proofreading and possible language editing would improve readability.

Author Response

Reviewer #1

General Comments:

This manuscript provides a retrospective cohort study assessing the effect of intraoperative opioid dosage and type of anesthesia on perioperative and long-term outcomes in patients undergoing radical cystectomy (RC) for urothelial carcinoma of the bladder. Results demonstrate that intraoperative opioids did not adversely impact survival rates or recurrence rates; combined intravenous and epidural anesthesia proved more successful at improving overall survival with decreased ICU admissions and ICU stays.

The topic is clinically significant, as understanding the impact of anesthetic management on oncological outcomes could inform perioperative practices. Overall, the manuscript is well structured and written; however, there may be several methodological issues or areas needing clarification that need addressing in order to strengthen validity and impact of this research study.

Specific Comments:

Abstract:

- Clarity: Abstract is brief yet could more effectively reflect limitations of study. Consider adding a sentence acknowledging retrodesign and calling for further research to ensure accuracy.

We thank Reviewer 1 for the proposal. Retrospective design of the study as well as the need for further research were now acknowledged in the abstract.

- Results Presentation: Note the nature of any Hazard Ratios (HRs) or Odds Ratios (ORs) reported. For example, adjusted or unadjusted should be stated accordingly.

We thank Reviewer 1 for the suggestion. All HRs and ORs were unadjusted based on the univariate regression analysis. This was added in the abstract.

Introduction:

- Background Rationale: The introduction provides an effective overview of RC and opioids' potential impact on cancer outcomes, but could use additional detail about how opioids might influence tumor progression; such as through immunosuppression or angiogenesis.

We thank Reviewer 1 for the proposal. This was added in the introduction.

- Literature Context: Exploration of discrepant findings from previous studies regarding intraoperative opioid usage for various cancer types will help establish a clearer rationale for this current research study.

We thank Reviewer 1 for the suggestion. The literature context was now expanded accordingly.

- Aim Clarification: The study objectives have been stated, but could be further refined. Be clear as to whether your focus lies with overall survival, recurrence rates, postoperative complications or all these outcomes.

We thank Reviewer 1 for the comment. The aim of the present study was now clearly defined.

Methods:

- Study Design:

- Retrospective Nature: At the start of your Methods section, clearly state that this study is retrospective cohort in nature.

We thank Reviewer 1 for the suggestion. We clearly state in the methodology that we performed a retrospective study.

- Patient Selection:

  - Inclusion/Exclusion Criteria: Provide specific inclusion and exclusion criteria. For instance, include information regarding how mixed histologies were managed as well as whether patients receiving either neoadjuvant or adjuvant chemotherapy are considered.

We thank Reviewer 1 for the question. Patients with mixed histologies were excluded, and neoadjuvant or adjuvant chemotherapy was not an exclusion criterion.

  - Baseline Differences: Explain how any possible baseline differences were addressed between anesthesia groups.

We thank Reviewer 1 for the question. The two groups presented similar baseline characteristics. Therefore, no further adjustments in the form of a multivariate analysis, or a matched-pared analysis were performed.

- Anesthesia Protocol:

  - Decision Process: Provide details regarding how the decision between intravenous only or combined intravenous and epidural anesthesia was reached; was it determined based on patient preferences, surgeon preferences or anesthesiologist discretion?

We thank Reviewer 1 for the valuable question. Combined intravenous and epidural anesthesia was discussed with all eligible patients by the anesthesiologists aiming for shared decision-making. The latter was amended accordingly.

  - Standardization: Indicate whether dosing of epidural sufentanil was standard across patients.

We thank Reviewer 1 for the query. Indeed, sufentanil was the only opioid substance applied for epidural anesthesia with a standardized approach based on relevant anesthesiological protocols. This was added.

  - Opioid Conversion: Provide more detail on how remifentanil and sufentanil doses were converted to morphine milligram equivalents (MMEs), including any references or formulae used.

We thank Reviewer 1 for bringing up this issue. Based on available literature (see references 16 and 17), 10 MME was considered the equivalent of 50 μg of i.v. remifentanil, which has approximately the same therapeutic potency as fentanyl, and 5 μg of i.v. sufentanil, which is about ten times as potent as fentanyl. In other words, 10 MMEs were equal to 50 μg of i.v. remifentanil and 5 μg of i.v. sufentanil. Based on the previous notion, 5 μg of i.v. remifentanil were equal to 1 MME and 0.5 μg of i.v. sufentanil were also equal to 1 MME.

- Outcomes and Definitions:

  - Primary and Secondary Outcomes: Define both primary and secondary outcomes clearly. For instance, outline whether overall survival refers to all-cause mortality or cancer-specific mortality.

We thank Reviewer 1 for the proposal. The primary outcome was clearly defined.

  - Perioperative Outcomes: Define what constitutes a serious Clavien-Dindo complication and how ICU admission decisions were determined (e.g., predefined criteria or clinician judgment).

We thank Reviewer 1 for the proposal. All Clavien-Dindo complications were defined based on the validated criteria for urological operations. Moreover, the need for admission to the ICU was based on clinical judgment according to the preoperative patient characteristics as well as the intraoperative findings.

- Statistical Analysis:

  - Adjustment for Confounders: Detail any methods used to adjust for potential confounding variables, such as multivariable regression analyses.

We thank Reviewer 1 for the suggestion. Unfortunately, the relatively low number of included patients and events did not permit us to perform multivariate regression analyses to adjust for further perioperative characteristics that might have affected outcomes. This is underlined in the limitations sections.

  - Rationale for Cut-offs: Justify the choice of the 75th percentile as the cut-off for high versus low opioid dosage groups.

We thank Reviewer 1 for the question. The rationale for cut-offs was based on clinical judgment. More specifically, we aimed to assess the effect of high opioid dosage since higher opioid dosages have been associated with worse oncological outcomes. This was further discussed now both in the methods section and in the discussion.

 - Handling of Missing Data: Explain how missing data were addressed, if applicable.

We thank Reviewer 1 for the question. As stated in the selection criteria in the methods, only patients with complete demographic data, full documentation of intraoperative opioid doses, and comprehensive follow-up information were included in the study.

  - Software Version: Update the statistical software version if more recent than R 3.6.3 is available, or justify using this version.

We thank Reviewer 1 for the question. The version of 3.6.3 is an actual R version and it is used for all analyses performed by our research group.

Results:

- Baseline Characteristics:

  - Comparative Analysis: Provide p-values for all comparisons in Table 1 to highlight any significant differences between the anesthesia groups.

We thank Reviewer 1 for the proposal. P-values are provided for all comparisons in Table 1 and statistically significant p-values are mentioned with bold. The latter was added to all relevant tables.

  - Additional Variables: Consider including other relevant variables, such as ASA scores, comorbidities, or tumor characteristics, and discuss any significant differences.

We thank Reviewer 1 for the proposal. Nevertheless, these relevant variables have already been provided in Table 1. Due to the extended manuscript, it was preferred to be discussed only in the tables section and not in the main text.

- Survival and Recurrence Outcomes:

  - Follow-up Duration: The median follow-up of 270 days is relatively short for oncological outcomes. Acknowledge this limitation in both the Results and Discussion sections.

We thank Reviewer 1 for bringing up this issue. The relatively short follow-up was added as an additional limitation of the present study.

  - Multivariable Analysis: Given the number of events (108 deaths and 106 recurrences), it seems feasible to perform multivariable analyses to adjust for confounders. Explain why this was not done or include these analyses.

We thank Reviewer 1 for the question. A multivariable analysis could not be performed due to the low number of events in the group of intravenous anesthesia only. In particular, 26 deaths and 16 recurrences were reported in patients undergoing only intravenous anesthesia. Given that based on the rule of thumb a minimum of 10 events per group is needed to adjust for further risk factors, and considering that we had 16 and 26 events, we preferred to limit our analyses to a univariate regression model.

  - Figures: Ensure that Figure 1 (Kaplan-Meier curve) includes clearly labeled axes, numbers at risk, and confidence intervals if possible.

We thank Reviewer 1 for the proposal. Labeled axes and numbers at risk are provided. Nevertheless, only the p-value is provided and not the confidence intervals to keep the figure simple and facilitate readers. Accordingly, all confidence intervals for the regression analyses are provided in the main text.

- Perioperative Outcomes:

  - ICU Admissions: Discuss possible reasons why combined anesthesia was associated with lower ICU admission rates.

We thank Reviewer 1 for the question. Lower ICU admission rates after combined anesthesia might be attributed to better perioperative pain control and improved respiratory function, leading, in turn, to better hemodynamic stability and reduction in systemic inflammatory response. This was amended in the first paragraph of the discussion.

  - Complication Rates: Provide more detail on the types of complications observed and whether they differed between groups.

We thank Reviewer 1 for the proposal. All complication rates of interest are depicted in Tables 3 and 4.

Discussion:

- Interpretation of Findings:

  - Mechanisms: Explore potential mechanisms for the observed association between combined anesthesia and improved survival and reduced ICU admissions.

We thank Reviewer 1 for the suggestion. The latter is now thoroughly discussed. Please see the other comments for further details.

  - Comparison with Literature: Expand the discussion to compare and contrast your findings with those of previous studies, such as the study by Chipollini et al. mentioned in the text.

We thank Reviewer 1 for the proposal. The findings of Chipollini et al. were now discussed and compared to the findings of our study.

- Limitations:

  - Acknowledgment: Discuss the retrospective design, potential selection bias, lack of randomization, and the short follow-up period as limitations.

We thank Reviewer 1 for the suggestion. All these limitations were now acknowledged in the limitation section.

  - Confounding Factors: Address the possibility of unmeasured confounders influencing the results, such as variations in surgical technique or perioperative care.

We thank Reviewer 1 for the proposal. The latter was also added to the limitations.

- Recommendations:

  - Future Research: Suggest the need for prospective, multicenter studies or randomized controlled trials to confirm these findings.

We thank Reviewer 1 for the suggestion. The latter was added into the conclusion.

Conclusion:

- Balanced Summary: The conclusion should reflect both the findings and the limitations of the study. Emphasize that while intraoperative opioid use did not adversely affect outcomes, the observational nature of the study warrants cautious interpretation.

We thank Reviewer 1 for the suggestion. The latter was added into the conclusions.

Additional Comments:

- Ethical Considerations:

  - Ethics Statement: The ethics statement is contradictory. It mentions approval by the Ethics Committee but also states that the article does not contain any studies with human participants. Clarify this section to accurately reflect the ethical approval and consent processes.

We thank Reviewer 1 for bringing up this issue and apologize for the typo. The ethics statement was modified accordingly.

- Data Availability:

  - Transparency: If possible, provide information on how the dataset can be accessed for verification or further research, respecting patient confidentiality.

We thank Reviewer 1 for the question. All data generated or analyzed during this study are included in this article. Further inquiries can be directed to the corresponding author.

Reviewer 2 Report

Comments and Suggestions for Authors

Thank you for submitting your work to this journal.

I would kindly recommend to modify the content of the Study design chapter, which I consider misleading. Even if your database is prospectively maintained, the study is still retrospective. Your patients most likely signed an informed consent form at the time of their initial admission into the hospital and not when they were included into this study.

The last paragraph of the introduction states that your aim was to assess the risk of higher intraoperative opioid doses on the prognosis of the patients while in the subchapter entitled Outcomes you state that your primary endpoint was to asses the role of opioids in predicting the perioperative outcomes. While the two objectives are similar, they are not identical and need to be conciliated.

Speaking about limits, the different types of urinary diversion are a significant factor when it comes to prognosis and outcomes, I would speculate it actually is more significant that the type of anesthesia. Same for the number of surgeons involved.

Please read and cite the following paper on a similar topic:

Persu C, Ciofu I, Petrescu A, Chirca N, Cauni V. Bladder Wall Structure Alterations in Patients Treated With Botulinum Toxin for Detrusor Overactivity - A Morphological Study. In Vivo. 2023 Mar-Apr;37(2):898-903. doi: 10.21873/invivo.13159. PMID: 36881062; PMCID: PMC10026681.

I would gladly revise an updated version of your paper.

Author Response

Reviewer #2

Thank you for submitting your work to this journal.

I would kindly recommend to modify the content of the Study design chapter, which I consider misleading. Even if your database is prospectively maintained, the study is still retrospective. Your patients most likely signed an informed consent form at the time of their initial admission into the hospital and not when they were included into this study.

We thank Reviewer 2 for the proposal. It is now clearly stated that we performed a retrospective analysis to avoid misinterpretations.

The last paragraph of the introduction states that your aim was to assess the risk of higher intraoperative opioid doses on the prognosis of the patients while in the subchapter entitled Outcomes you state that your primary endpoint was to asses the role of opioids in predicting the perioperative outcomes. While the two objectives are similar, they are not identical and need to be conciliated.

We thank Reviewer 2 for the suggestion. The aims of the study were now clearly written to capture overall survival, recurrence rates, and postoperative complications.

Speaking about limits, the different types of urinary diversion are a significant factor when it comes to prognosis and outcomes, I would speculate it actually is more significant that the type of anesthesia. Same for the number of surgeons involved.

We thank Reviewer 2 for bringing up this issue. Even though, the groups did not differ in the type of urinary diversion performed, the different types of urinary diversion as well as the surgeon’s caseload might have affected the outcomes. Still, it should be highlighted that it was beyond the scope of the present study to assess for these risk factors since multiple studies on the matter have been performed either from our research group or from other research groups. Thus, this was mentioned as an additional limitation of the present study.

Please read and cite the following paper on a similar topic:

Persu C, Ciofu I, Petrescu A, Chirca N, Cauni V. Bladder Wall Structure Alterations in Patients Treated With Botulinum Toxin for Detrusor Overactivity - A Morphological Study. In Vivo. 2023 Mar-Apr;37(2):898-903. doi: 10.21873/invivo.13159. PMID: 36881062; PMCID: PMC10026681.

We thank Reviewer 2 for the proposal. This interesting study was now cited.

Round 2

Reviewer 1 Report

Comments and Suggestions for Authors

I confirm that the authors have responded to all comments and revised the article accordingly, making it suitable for publication.

Reviewer 2 Report

Comments and Suggestions for Authors

Thank you for making the changes I suggested.